# Noninvasive Biomarkers of Human Embryo Developmental Potential

**DOI:** 10.3390/ijms26104928

**Published:** 2025-05-21

**Authors:** Jan Tesarik

**Affiliations:** MARGen (Molecular Assisted Reproduction and Genetics) Clinic, Calle Gracia, 36, 18002 Granada, Spain; jantesarik13@gmail.com; Tel.: +34-606376992

**Keywords:** human embryo, preimplantation development, in vitro fertilization, single-embryo transfer, noninvasive biomarkers

## Abstract

There are two types of noninvasive biomarkers of human embryo developmental potential: those based on a direct assessment of embryo morphology over time and those using spent media after embryo in vitro culture as source of information. Both are derived from previously acquired knowledge on different aspects of pre-implantation embryo development. These aspects include embryo morphology and kinetics, chromosomal ploidy status, metabolism, and embryonic gene transcription, translation, and expression. As to the direct assessment of morphology and kinetics, pertinent data can be obtained by analyzing sequential microscopic images of in vitro cultured embryos. Spent media can serve a source of genomic, metabolomic, transcriptomic and proteomic markers. Methods used in the early pioneering studies, such as microscopy, fluorescence in situ hybridization, autoradiography, electrophoresis and immunoblotting, or enzyme-linked immunosorbent assay, are too subjective, invasive, and/or time-consuming. As such, they are unsuitable for the current in vitro fertilization (IVF) practice, which needs objective, rapid, and noninvasive selection of the best embryo for uterine transfer or cryopreservation. This has been made possible by the use of high-throughput techniques such as time-lapse (for direct embryo evaluation), next-generation sequencing, quantitative real-time polymerase chain reaction, high-performance liquid chromatography, nanoparticle tracking analysis, flow cytometry, mass spectroscopy, Raman spectroscopy, near-infrared spectroscopy, and nuclear magnetic resonance spectroscopy (for spent culture media analysis). In this review, individual markers are presented systematically, with each marker’s history and current status, including available methodologies, strengths, and limitations, so as to make the essential information accessible to all health professionals, even those whose expertise in the matter is limited.

## 1. Introduction

More than forty years after the birth of the first baby conceived by in vitro fertilization (IVF) [1], many challenges to improving IVF efficacy and safety still remain [2]. In fact, the overall IVF success has not augmented much above 30%, although the conditions for embryo culture have improved [3]. Success (term pregnancy) after IVF basically depends on two types of factors: those related to uterine receptivity and those given by embryo developmental potential [4]. Embryo developmental potential has been addressed by a number of studies aimed at improving IVF efficiency. However, most of the initial studies used techniques incompatible with their use in the clinical IVF settings because of issues related to invasiveness and subjectivity. These limitations created a need for fast, objective, and noninvasive techniques to be incorporated into the decision-making scheme used in IVF centers, especially with regard to single-embryo transfer. To meet this need, new techniques based on computer-assisted processing of sequential static embryo images and molecular analyses of spent media from embryo culture were developed [5]. The aim of this paper is to review the history, state of the art, and strengths and limitations of noninvasive embryo-quality biomarkers currently available for use in IVF laboratories.

## 2. Noninvasive Biomarkers: History, Current State, and Strengths and Limitations

Repeated observations on zygote and embryo morphology at different time points after fertilization are the source of many noninvasive biomarkers aiding in prediction of embryo developmental potential [6,7,8,9,10,11,12,13,14,15,16,17]. Experience showing that in many cases such morphology-based approaches cannot detect functional deficiencies, such as chromosomal abnormalities and issues related to gene expression or metabolic activity, was at the origin of research into the suitability of spent media from embryo culture as a source of information. This eventually led to the development of a variety of noninvasive “omics” methods currently available (Figure 1). These methods serve for the evaluation of embryo ploidy status (genomics), gene transcription (transcriptomics) and translation (proteomics and secretomics), and the metabolism of different substrates (metabolomics) [5]. In the beginning, embryos were cryopreserved (vitrified) until the test result was available and transferred to the uterus later (Figure 1).

Nowadays, increase in the speed of “omics” techniques has made it possible to transfer fresh, marker-selected embryos in the same treatment cycle. The history, current status, and strengths and limitations of each category of noninvasive biomarkers are detailed below.

### 2.1. Morphology and Kinetics

#### 2.1.1. History

From the very beginning of human in vitro fertilization (IVF), there have been attempts at using static images of the zygote and the embryo at successive stages of preimplantation development (Table 1) as the basis for quality assessment of individual embryos with regard to their presumed developmental potential [18,19]. Zygotes were graded based on the distribution of nucleolar precursor bodies (NPBs) in both pronuclei (PN) at the time of pronuclear apposition [8], and it was shown that the “pronuclear pattern” (Figure 2) has a high predictive value as to further embryo development and chromosomal status [20]. As for cleaving embryos, they were assessed by the number and symmetry of blastomeres, the grade of fragmentation, the eventual presence of multinucleated blastomeres, and the compaction status [21]. Blastocysts were scored according to the degree of blastocoel expansion and the definition of the first two cell lineages, the inner cell mass, and the trophectoderm [9]. However, in spite of some encouraging reports, morphological grading of cleaving embryos and blastocysts was shown to exhibit a poor correlation with the success of IVF [22], probably related to subjectivity due to intra- and interobserver variability [23] (Table 1).

In order to eliminate the subjectivity bias, systems based on automated assessment of embryo morphology over time, with the use of time-lapse technology and morphokinetic algorithms employed for time-lapse data interpretation, were introduced into IVF technology during the first decade of this century [16].

#### 2.1.2. Current Status

Several markers of embryonic kinetics were reviewed recently, and some of them have been applied in decisional algorithms for embryo transfer [24]. Out of 35 morphometric, morphologic, and morphokinetic variables, only the location of PN in the one-cell zygote and the absence of multinucleated blastomeres (MNB) at the two-cell stage were linked to live birth rate. Specifically, central PN juxtaposition was associated with a two-fold increase in the odds of live birth, whereas the presence of MNB was linked to half the odds of live birth, and these two parameters were independent of embryo kinetics [24]. Morphological criteria were also applied to frozen–thawed blastocysts, and it was shown that blastocyst re-expansion rate shortly (9–11 min) after warming, assessed with the use of image-analysis software, is a strong dynamic indicator of embryo quality [25].

Recently, it was attempted to enhance the predictive power of criteria based on static morphologic evaluation of embryo images by using artificial intelligence (machine learning) to create virtual 3-dimensional (3D) images from time-lapse recordings (time-lapse 3D holotomography), and impressive performance metrics with regard to blastocyst development were reported [26]. However, a pilot randomized controlled trial (RCT) failed to detect any significant difference in a multicenter, randomized, double-blind noninferiority parallel-group trial conducted across 14 IVF clinics in Europe and Australia, assessing the value of deep learning in selecting the optimal embryo for transfer; it was not able to demonstrate noninferiority of deep learning for clinical pregnancy rate when compared to standard morphology evaluation [27]. This conclusion was confirmed by another independent RCT [28]. Consequently, despite being promising, the time-lapse technology still needs to be perfected, possibly by incorporating PN morphology scoring (Figure 2), which was previously shown to have a high predictive potential for further embryo development, embryo ploidy, and IVF outcome [8,20]. Even so, time-lapse is evidently more rapid and embryo-friendly than the observer-operated grading since it avoids prolonged periods of stay of embryos outside the incubator. In addition to morphology, the spatiotemporal organization of the first mitotic division, analyzed with the use of a unique time-lapse dataset, was shown to predict pregnancy after embryo transfer and to be related to the zygote PN pattern [29].

Beyond embryo viability assessment in general, another recent study [30] reported the development of an efficient artificial intelligence model that can predict embryo ploidy status from time-lapse recordings, showing an accuracy of 0.74, an aneuploid precision of 0.83, and an aneuploid predictive value (recall) of 0.84 [31]. Moreover, relevant information can be obtained by time-lapse analysis of embryos as early as the beginning of the first cleavage division [32].

**Table 1 ijms-26-04928-t001:** Overview of the most important information concerning biomarkers based on embryo morphology and kinetics. Abbreviations: MNB = multinucleated blastomeres; 3D = three-dimensional.

Marker	Reference Outcome	Predictive Capacity	Techniques	References
**Pronuclear**	Ongoing development	Very high	Single microscopic	[8]
**morphology**	Chromosomal ploidy	Very high	observation	[20]
**Cleavage speed and regularity**	Pregnancy	Low	Repeated microscopic observation	[21,22,23]
**Proportion of MNB**	Developmental arrest and pregnancy failure	High	Repeated microscopic observation	[24]
**Noninvasive continuous recording**	Developmental arrest and pregnancy failureChromosomal ploidy	Debated High with 3D setting (holotomography)High	Time-lapse photography	[25,26,27,28,29] [30,31,32]

#### 2.1.3. Strengths and Limitations

Unlike some other noninvasive biomarkers of embryo quality (see below), those based on morphology and kinetics can be applied to embryos from the very early stages, including the one-cell zygote. Good correlations with further embryo development and clinical IVF outcomes were reported for some of them in early observational studies. Evaluation using time-lapse technology eliminates the bias of subjectivity and is faster and more embryo-friendly, but its superiority over the classical observer-operated grading systems has not been confirmed by the only two RCTs published so far. Morphological grading does not require costly equipment when realized in an observer-operated manner. For generation and processing of time-lapse images, the purchase of more expensive equipment is needed.

### 2.2. Chromosomal Ploidy Status (Genomics)

#### 2.2.1. History

Aneuploidy is a well-known condition underlying implantation failure, miscarriage, and abortion. It mainly arises from meiotic errors in oocytes, closely related to maternal age [33], although it can also be caused by sperm abnormalities and mitotic errors during the early cleavage divisions [34]. When caused by oocyte meiotic errors, individual embryonic cells usually exhibit a uniform aneuploidy pattern [35], while chromosomally mosaic embryos, containing both euploid and aneuploid cells, typically result from mitotic errors during embryo cleavage [36]. To cope with the aneuploidy issue, preimplantation genetic testing for aneuploidies (PGT-A) was adopted into IVF practice starting in the early 1990s [37].

PGT-A was first used by analyzing cells of abnormally developing embryos by fluorescence in situ hybridization (FISH) with specific probes for a limited number of chromosomes. The five chromosomes most involved in human aneuploidy (X, Y, 18, 13, and 21) were assessed in the original study [37]. In order to maintain the viability of the tested embryo, trophectoderm biopsy, a micromanipulation technique whereby a small group of cells are extracted from blastocyst trophectoderm [38], was subsequently used for PGT-A in the clinical IVF setting. To extend PGT-A to more or all chromosomes, FISH was progressively substituted with still more effective techniques, such as polymerase chain reaction (PCR), comparative genomic hybridization (CGH), and next-generation sequencing (NGS), and trophectoderm biopsy techniques were also refined [39]. However, recent findings challenged both the reliability and the harmlessness of PGT-A using trophectoderm biopsy, even with the use of the most advanced molecular techniques, putting PGT-A in its traditional form into question (Table 2). Most criticisms concerned the representativity of randomly sampled trophectoderm cells as to the chromosomal status in cells of the inner cell mass, the potential harm trophectoderm biopsy may cause to the embryo, and the issue of mosaic embryos.

Trophectoderm cells contribute to the placenta, while cells of the future embryonic and fetal body stem from the inner cell mass. This creates a probability that the biopsied cells will not hold genetic material that represents the embryo’s genetic material [40]. Harmful effects of trophectoderm biopsy on embryo developmental potential were reported [41], and this micromanipulation might also increase the incidence of mosaic blastocysts [42]. Finally, with the use of trophectoderm cells, the PGT-A result is poorly predictive of the absolute level of mosaicism of a single embryo as demonstrated with the use of computational modeling [43].

To overcome the above problems, alternative methods for embryo ploidy assessment by analyzing samples of spent embryo-culture media containing soluble DNA released from the embryo (Figure 3) were sought, resulting in the development of noninvasive chromosome screening (NICS) [44]. NICS, also referred to as noninvasive PGT-A (niPGT-A), makes use of NGS after multiple annealing and looping-based amplification cycles (MALBAC) for whole-genome amplification (WGA) [44] and is highly recommended for embryo ploidy assessment (Table 2).

#### 2.2.2. Current Status

niPGT-A, a kind of noninvasive liquid biopsy used in other fields of medicine (Figure 3), was reported to exhibit high sensitivity, specificity, positive predictive value, and negative predictive value, with low false-positive and false-negative rates, comparing favorably to classical PGT-A in terms of ploidy diagnostic accuracy [45]. Recently, niPGT-A was successfully used in a single-embryo transfer setting to select euploid embryos in patients carrying balanced chromosomal rearrangements by using genome-wide single nucleotide polymorphism genotyping and haplotyping approach [46], and algorithms for minimizing the impact of media contamination with maternal DNA were developed and validated [47]. In spite of these encouraging findings, the classical PGT-A, performed on whole surplus IVF embryos donated for research, still remains a gold standard for validation of new noninvasive biomarkers. In parallel to PGT-A and niPGT-A, there is growing interest in using morphokinetic embryo analysis (see Section 2.1), alone or in combination with other embryonic and patient parameters, to replace both of them. Notably, the development of a noninvasive artificial intelligence approach for the prediction of human blastocyst ploidy, based on embryo morphology, morphokinetics, and associated clinical information, was reported recently [48].

#### 2.2.3. Strengths and Limitations

Both traditional PGT-A and niPGT-A can provide information on embryo developmental potential and thus help select the best embryo for elective transfer. However, traditional PGT-A is burdened with errors due to the interpretability of findings obtained with a few trophectoderm cells for the whole embryo, while more RCTs are still needed to establish niPGT-A as the preferential method. Results can be obtained faster with niPGT-A than with PGT-A, but both approaches require expensive equipment and skilled staff. Indirect evaluation methods based on synthesis of morphological and morphokinetic data, sometimes also including clinical data of the patients, may be the best solution in the future.

### 2.3. Gene Activity

The initial stages (at least the first two cell cycles) of human embryo development rely almost completely on molecules (maternal mRNA and proteins) stored in the oocyte, and oocyte-derived message continues to be used for some time even after the beginning of embryonic gene activity [49]. Major embryonic gene activation occurs between the four-cell and the eight-cell stage of development, later that in most laboratory animal species [50], even though low levels of gene expression can be detected as early as the one-cell stage [51,52,53]. However, products of the minute gene activity before the human embryo achieves the four-cell stage can hardly be expected to be detectable in spent embryo-culture media. The major activation of RNA synthesis (transcription) and protein synthesis on newly synthesized embryonic RNA templates (translation) are milestones in embryo development and underlie the possibility of using transcriptomics, proteomics, and secretomics in the search for specific biomarkers to distinguish between embryos that have accomplished these developmental event and those that have not.

#### 2.3.1. Transcription (Transcriptomics)

##### History

After a near-silent period, a major transcriptional activation was detected in human embryos at the four-cell stage [54]. Because this event only occurs in morphologically normal blastomeres and is severely disturbed in abnormal (such as multinucleated) ones [55] and because some of the RNA species synthesized by healthy blastomeres are secreted into and are stable in embryo culture media, attempts were made to use specific embryo-secreted RNAs or combinations thereof as embryo-quality markers [56,57,58]. There is a great variety of RNA types, differing from each other by size, main site of intracellular localization, and function. Special attention has been paid to non-coding RNAs (ncRNAs), which represent as much as 98% of the transcriptome, and some types of them, mainly microRNAs (miRNAs) and piwi-interacting RNAs (piRNAs), were shown to be related to embryo viability and developmental potential [56,57,58].

MiRNAs are composed of 18–24 nucleotides and act as gene expression modulators through translational inhibition or degradation of messenger RNAs (mRNA) [59]. They were suggested to regulate adhesion of the blastocyst to the endometrium at the outset of implantation [60]. For 10 years, microRNAs, secreted from human blastocysts into culture media, have been considered potential biomarkers for noninvasive embryo selection [60,61,62,63,64] and for predicting the risk of biochemical pregnancy loss after embryo transfer [65]. Another type of snRNAs, piRNA, is involved in gene expression regulation and genome protection from instability and has also been suggested as an embryo biomarker [56] (Table 3).

##### Current Status

A recent study using massive-parallel sequencing on the Illumina sequencing platform and the BioMAI Pipeline Predictor Model specifically identified two miRNAs (miR-16-5p and miR-92a-3p) and five piRNAs (piR-28263, piR-18682, piR-23020, piR-414, and piR-27485) that prognostically and predictively distinguished a high-quality embryo suitable for IVF transmission from a low-quality embryo, with high sensitivity, specificity, and accuracy [56] (Table 3).

In another study analyzing the spent culture medium of day 3 and day 5 embryos and quantitative real-time polymerase chain reaction (qRT-PCR), three miRNAs, including hsa-miR-199a-5p, hsa-miR-483-5p, and hsa-miR-432-5p, were shown to be correlated with pregnancy and proposed as biomarkers for embryo quality during IVF cycles [66]. A recent analysis using machine learning models also revealed different gene expression patterns between euploid blastocysts that gave a clinical pregnancy and those that did not [67].

##### Strengths and Limitations

Embryo evaluation employing a panel of ncRNA markers with the use of qRT-PCR techniques is rapid and highly sensitive and thus advantageous for fast embryo assessment in clinical IVF practice. On the other hand, it requires appropriately skilled staff and relatively expensive equipment.

#### 2.3.2. Translation (Proteomics and Secretomics)

##### History

A major activation of gene translation was detected in human embryos at the eight-cell stage [68]. At the same stage, the first morphological signs of embryonic gene expression were found with the use of quantitative ultrastructural analysis [69]. The proteome represents all proteins translated from specific gene expression products of a cell at a specific time, while the secretome refers to the proteins produced and secreted by the developing embryo [5].

**Table 3 ijms-26-04928-t003:** Overview of the most important information concerning biomarkers based on embryo gene activity. Abbreviations: miRNAs, microRNAs; piRNAs, piwi-interacting RNAs; qRT-PCR, quantitative real-time polymerase chain reaction; ELISA, enzyme-linked immunosorbent assay.

Marker	Reference Outcome	Predictive Capacity	Techniques	References
**Transcriptome** miRNAs piRNAs	Embryo viability and pregnancy	Very high	Massive-parallel sequencing, qRT-PCR	[56,57,58,59,60,61,62,63,64,65,66,67]
**Proteome**	Embryo viability, ploidy, and pregnancy	Very high	Historical: ELISA, Western blotting	[70]
			Current: Mass spectrometry	[71,72,73,74,75,76,77]

##### Current Status

Both proteins secreted into the media (increasing trend) and those already present and taken up by the embryo (decreasing trend) can be valuable biomarkers of embryo development [70]. Relatively simple techniques, such as ELISA and Western blotting, can be used to assess proteomics of spent embryo culture media. However, the use of high-throughput methodologies, such as mass spectrometry, eventually enhanced by matrix-assisted laser desorption ionization–time of flight mass spectrometry (MALDI-ToF MS) [71], can make the analysis faster and more sensitive (Table 3). By using MALDI-ToF MS to compare euploid and aneuploid embryos (tested by PGT-A), it was possible to characterize 12 unique spectral regions for euploid embryos and 17 for aneuploid ones [72], and the technique showed a high positive predictive value for ongoing pregnancy after single-embryo transfer [73].

Individual specific markers of interest include human chorionic gonadotropin (hCG) isoforms [74], interleukin (IL) 6, stem cell factor (SCF), and interferon (IFN) α2 [75], along with soluble human leukocyte antigen G (sHLA-G) [76] and soluble cluster of differentiation 146 (sCD146) [77], the latter being correlated with IVF success negatively. Moreover, another 18 exclusively expressed proteins were identified in the positive implantation group and 11 in the negative embryo implantation group [77].

##### Strengths and Limitations

Current noninvasive techniques available for embryonic proteome and secretome assessment require as low as 1 μL of the sample volume, do not require the acquisition of special skills by the clinical laboratory staff, and represent ultra-fast tools for embryo selection immediately prior to uterine transfer. On the other hand, a general agreement on which technique/marker is preferrable still remains to be reached, as results can be influenced by the composition of embryo culture media and thus difficult to extrapolate from one laboratory setting to another, and the purchase of relatively expensive equipment is required.

### 2.4. Substrate Uptake and Secretion (Metabolomics)

#### 2.4.1. History

It has been known for many years that mammalian embryos display a rapid qualitative change in their metabolic activity at a certain point of preimplantation development [78]. This was first described for energy metabolism. In fact, both mouse [79] and human [80] embryos switch from pyruvate to glucose as the predominant energy source between compaction and blastulation, a period just after the major embryonic gene activation in humans (see Section 2.3 of this article). This switch is underlain by carbohydrate metabolism passing from the tricarboxylic acid (TCA) cycle to glycolysis [81]. In addition, patterns of the consumption and secretion of amino acids was also found to change in this period [82]. Based on these observations, depletion/appearance of components like carbohydrates in spent media from cleavage-stage embryo culture was suggested as a source of noninvasive markers of embryo developmental potential [83]. In earlier studies, data were generated by thin-layer chromatography (TLC), microfluorescence, or enzyme-linked immunosorbent assays (ELISA). At present, embryo metabolomics assessment preferentially employs high-performance liquid chromatography (HPLC) [84,85,86] and spectroscopy techniques that use energy changes to study the local environment of atoms, such as Raman, near-infrared (NIR), and nuclear magnetic resonance (NMR) spectroscopies [87] (Table 4).

#### 2.4.2. Current Status

When used in combination with morphology, Raman spectroscopy of day 3 embryo culture media gave promising results as to predicting embryo implantation potential [88]. More recently, a model based on Raman spectroscopy approach was reported to have a high specificity and sensitivity by receiver operating characteristics (ROC) analysis [89] (Table 4).

On the other hand, a double-blind RCT evaluating embryo selection by metabolomic profiling of day 3 embryo culture media by NIR plus morphology failed to detect a superior prediction power of this method as compared to morphology alone [90]. A similar conclusion was reported for NMR [91]. However, a recent study showed that Raman spectroscopy can achieve an accuracy rate of 71.5% by combining several machine learning methods and detected significant differences in tyrosine, tryptophan, and serine between spent media after culture of day 3 embryos that gave a pregnancy after transfer and those that did not [84,85,86,92].

#### 2.4.3. Strengths and Limitations

Currently available techniques are fast and do not require special skills of clinical laboratory personnel. However, due to differences in assay methods, sample size, sample collection, and statistical analysis methods, no single metabolomic assay has yet been accepted for clinical use worldwide. In addition, laboratory-specific conditions and factors like the type of culture medium [93,94] can all affect biomarker reliability.

### 2.5. Extracellular Vesicles

#### 2.5.1. History

Embryo-derived extracellular vesicles (EVs) are increasingly recognized as important mediators of embryo–maternal communication and carriers of bioactive molecules such as genomic DNA (gDNA) [95], RNAs (especially miRNAS) [96], and proteins [97]. They are increasingly recognized as important mediators of embryo–maternal communication, by exerting specific effects on information transportation, immune stimulation, and regulation of gene expression [97]. As early as 2017 embryo-derived EVs present in spent culture media were suggested as noninvasive biomarkers of embryo implantation potential [98].

#### 2.5.2. Current Status

Nanoparticle tracking analysis (NTA) and array comparative genomic hybridization (aCGH) analysis were performed to evaluate EVs and their genomic DNA (gDNA) content and showed a high predictive value for embryo chromosomal status and implantation [95]. Another study revealed that embryos leading to successful pregnancies secreted EVs with higher particle concentration, larger size, and distinct miRNA profiles compared to non-viable embryos [96]. By using propidium iodide to label the nucleic acid content of embryo-derived EVs followed by flow cytometry, it was possible to identify embryos that were most likely to implant before uterine transfer [99].

#### 2.5.3. Strengths and Limitations

EVs are very promising biomarkers of embryo implantation potential that can be assessed by simple, noninvasive, inexpensive, and quick tests. Further data, especially from RCTs, is needed to confirm the current expectations.

### 2.6. Combined Use of Different Biomarkers

As already marginally mentioned in Section 2 of this article, interesting data were obtained by combining data obtained by different biomarkers and integrating them into embryo development-predicting algorithms [100], and a model of the bioinformatics pipeline of artificial intelligence and machine learning, combining noninvasive biomarker data with sperm parameters, ovarian reserve, ovarian stimulation, uterine evaluation, patient genetics, and patient demographics, was suggested [56]. A human embryo model integrating an embryo single-cell RNA sequencing dataset covering development from the zygote to the gastrula was developed recently [101]. However, another recent report reveals that inflated expectations of new technologies, namely artificial intelligence-assisted embryo selection in IVF, are not justified by facts [102]. Therefore, while acknowledging the possibilities offered by such novel approaches, this is not the time to adopt too enthusiastic an expectancy.

## 3. Conclusions

Based on previous findings regarding morphology and kinetics, chromosomal status, gene activity, and metabolism of human preimplantation embryos, a number of distinct features or combination thereof were defined as potential noninvasive biomarkers of embryo developmental potential. As for morphology and kinetics, different extensions of time-lapse imaging technology were assessed. Regarding the rest, a number of “omics” methods have been designed. They include genomics, transcriptomics, proteomics, and metabolomics, reflecting embryo chromosomal ploidy status, transcription, translation, and metabolic activity, respectively. While no single method/marker has yet been accepted universally, this article reviews the current status and the strengths and limitations of each one of them so as to guide IVF centers in their choice of the most suitable method(s) with regard to their specific needs.

## Figures and Tables

**Figure 1 ijms-26-04928-f001:**
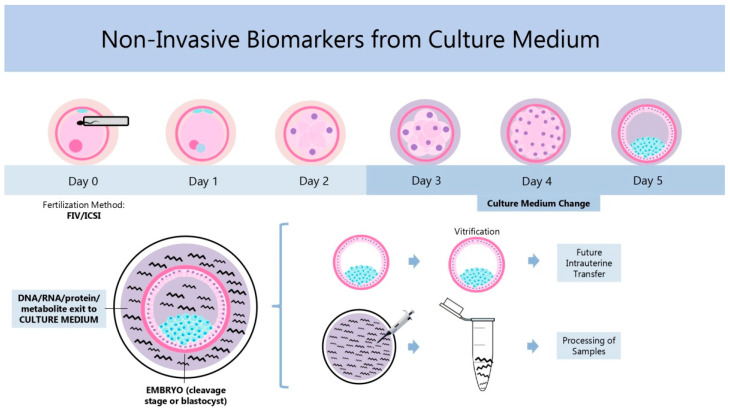
Schematic representation showing the timing of biological material sampling and processing for “omics” analyses. Between day 3 and day 5 post fertilization (depending on the type of test), samples of spent media from embryo culture containing embryo-derived DNA, RNA, protein, and metabolites are analyzed to select the best embryos for later embryo transfer or cryopreservation (vitrification).

**Figure 2 ijms-26-04928-f002:**
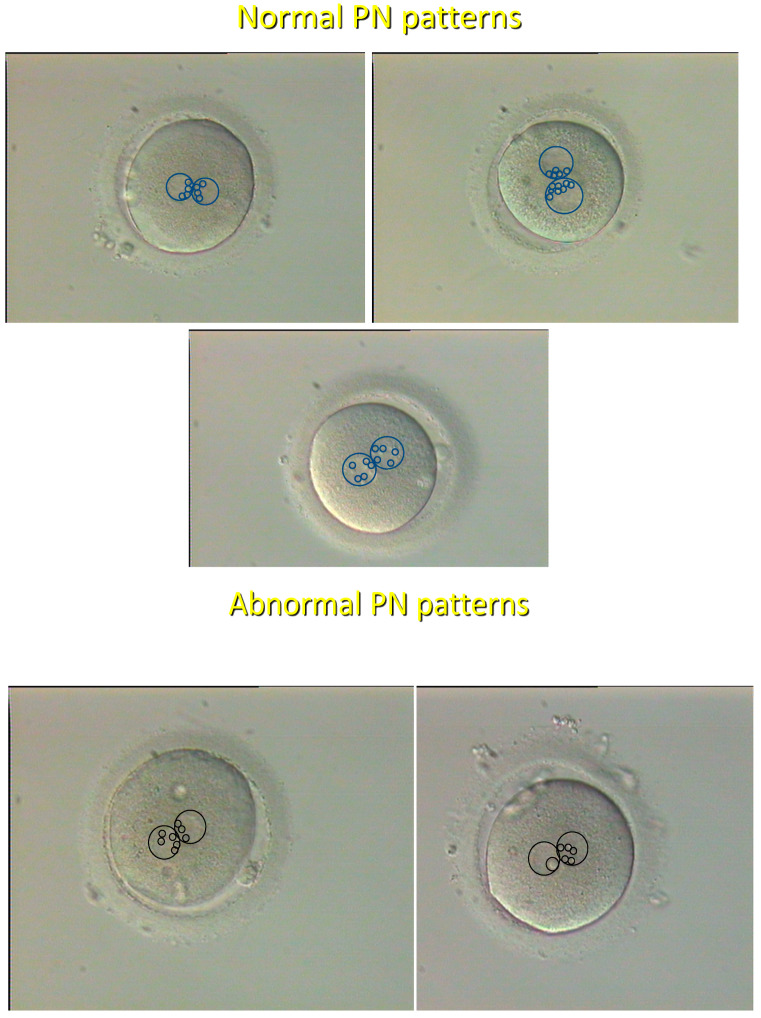
Normal (**upper** panel) and abnormal (**lower** panel) pronuclear (PN) patterns. Normal patterns are characterized by an equal size and distribution (dispersed or polarized) of nucleolar precursor bodies (NPBs) (small circles) in both pronuclei (large circles). Unequal size or distribution of NPBs is characteristic of abnormal patterns. See Tesarik and Greco [8] for proof of concept. Hoffman modulation contrast, original magnification ×200.

**Figure 3 ijms-26-04928-f003:**
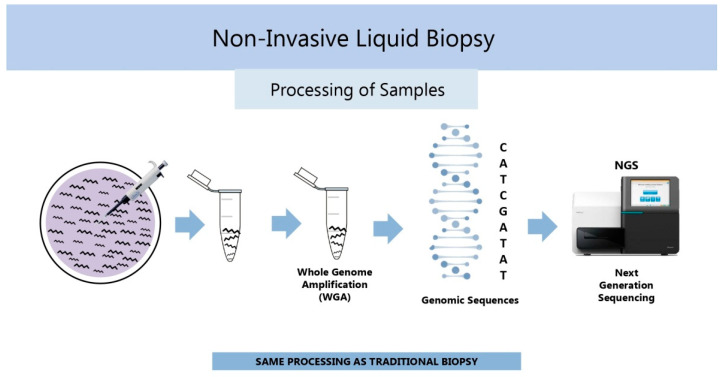
Schematic representation of spent embryo-culture media sampling and processing in noninvasive liquid biopsy for niPGT-A. From left to right: a sample of medium is taken from embryo culture dish, processed by WGA, and subjected to genomic analysis by next-generation sequencing.

**Table 2 ijms-26-04928-t002:** Overview of the most important information concerning biomarkers based on embryo chromosomal ploidy. Abbreviations: PGT-A, preimplantation genetic testing for aneuploidy; NICS, noninvasive chromosome screening; niPGT-A, noninvasive PGT-A; RCTs, randomized controlled trials, FISH, fluorescent in situ hybridization; PCR, polymerase chain reaction; CGH, comparative genome hybridization; NGS, next-generation sequencing; WGA, whole-genome amplification.

Marker	Reference Outcome	Predictive Capacity	Techniques	References
**PGT-A**	Whole-embryo ploidy and pregnancy	Intermediate and questionable	FISH, PCR, CGH, NGS	[37,38,39,40,41,42,43]
**NICS (niPGT-A)**	Whole-embryo ploidy and pregnancy	Very high but needing confirmation by RCTs	NGS after WGA	[44,45,46,47]

**Table 4 ijms-26-04928-t004:** Overview of the most important information concerning biomarkers based on embryo metabolome. Abbreviations: NIR, near-infrared; NMR, nuclear magnetic resonance; TLC, thin-layer chromatography; ELISA, enzyme-linked immunosorbent assay.

Marker	Reference Outcome	Predictive Capacity	Techniques	References
**Carbohydrate profile**	Embryo viability and pregnancy	Doubtful for NIR and NMR spectroscopies	Historical: TLC, ELISA Current: HPLC and Raman, NIR, or NMR spectroscopies	[83,84,85,86,87,88,89,90,91,92]
**Amino acid profile**	Embryo viability and pregnancy	High for HPLC and Raman spectroscopy		

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
