# Peer review of "Noninvasive Biomarkers of Human Embryo Developmental Potential"

_ijms, 2025, doi:10.3390/ijms26104928_

Round 1
Reviewer 1 Report
Comments and Suggestions for Authors
The manuscript presents a well-structured grounded synthesis of current non-invasive approaches for assessing embryo viability via analysis of spent embryo culture media (SECM). A key strength lies in its systematic categorization of transcriptomic, proteomic, and metabolomic approaches contextualized within the clinical challenges of embryo selection in assisted reproductive technologies (ART). The authors demonstrate a clear grasp of the limitations inherent to morphological and invasive genetic assessments, and compellingly argue for molecular alternatives that preserve embryo integrity while improving predictive accuracy.
Literature and recent technological advances (e.g., AI coupled time-lapse imaging, niPGT-A, Raman spectroscopy) are well-cited throughout. However, while the metabolomics section focuses primarily on Raman spectroscopy, near-infrared (NIR) spectroscopy, and nuclear magnetic resonance (NMR), the review does not mention High-Performance Liquid Chromatography (HPLC). Recent literature has shown that amino acid profiling using HPLC can significantly correlate with embryo quality, implantation rates, and pregnancy outcomes (Huo et al., 2020; Eldarov et al., 2022; Nami et al., 2024) Furthermore, the review could be strengthened by a clearer acknowledgment of the influence of laboratory-specific conditions on biomarker variability. Factors like the type of culture medium (Zagers et al., 2025) and how samples are handled — especially freezing and thawing — can all affect biomarker reliability, leading to potential misinterpretation of metabolic data (Cuhadar et al., 2013; An et al.,2021).
One area where the manuscript could be further strengthened is in its consideration of extracellular vesicles (EVs), which are increasingly recognized as important mediators of embryo-maternal communication and carriers of bioactive molecules such as RNAs, proteins, and genomic material.
Recent studies on human embryo-derived extracellular vesicles (EVs) from spent culture media have demonstrated their significant potential as non-invasive biomarkers for embryo quality and implantation success. Larger EVs correlated with higher embryo quality, although genomic DNA (gDNA) analysis revealed a high prevalence of chromosomal abnormalities (Veraguas et al., 2020). Comparative analysis indicated that embryos leading to successful pregnancies secreted EVs with higher particle concentration, larger size, and distinct miRNA profiles compared to non-viable embryos, positioning EV content as a promising prognostic tool (Pavani et al., 2022). Furthermore, lower counts of propidium iodide-positive (PI+) EVs in culture media strongly associated with implantation success, reinforcing the diagnostic value of EV nucleic acid content (Pallinger et al., 2017). EV-derived microRNAs (miRNAs) detected in SECM have been associated with pregnancy outcomes (Abu Halima, 2017).
It is also worth noting that the segment between lines 47 and 60 presents several generalized claims about non-invasive biomarkers and their clinical application without supporting references. Citing relevant literature here would reinforce the argument and strengthen the manuscript’s academic reliability.
In conclusion, the manuscript contributes meaningfully to the discussion of molecular approaches in ART, particularly in highlighting validated -omics tools and acknowledging the technical nuances of non-invasive embryo profiling.
Author Response
Dear Reviewer
Thank you very much for your comments which will certainly increase the manuscript quality. I have included the corresponding modifications in the manuscript (highligted). This is a list of the changes made.
Comment 1
Literature and recent technological advances (e.g., AI coupled time-lapse imaging, niPGT-A, Raman spectroscopy) are well-cited throughout. However, while the metabolomics section focuses primarily on Raman spectroscopy, near-infrared (NIR) spectroscopy, and nuclear magnetic resonance (NMR), the review does not mention High-Performance Liquid Chromatography (HPLC). Recent literature has shown that amino acid profiling using HPLC can significantly correlate with embryo quality, implantation rates, and pregnancy outcomes (Huo et al., 2020; Eldarov et al., 2022; Nami et al., 2024)
Response
A sentence about potential use of high-performance liquid chromatography (HPLC), together with the suggested references, has been added to subsection “History” of section 2.4. “Substrate Uptake and Secretion (Metabolomics)”. Further, in the subsection “Current Status” of the same section, the significant correlation between amino acid profiling by HPLC and embryo quality reported in the three recent studies has been highlighted.
Comment 2
Furthermore, the review could be strengthened by a clearer acknowledgment of the influence of laboratory-specific conditions on biomarker variability. Factors like the type of culture medium (Zagers et al., 2025) and how samples are handled — especially freezing and thawing — can all affect biomarker reliability, leading to potential misinterpretation of metabolic data (Cuhadar et al., 2013; An et al., 2021).
Response
A sentence stating that laboratory-specific conditions and factors like the composition of culture medium and sample handling, especially freezing and thawing, can all affect biomarker reliability, together with the suggested references, has been included in the subsection “Strengths and Limitations” of section 2.4. “Substrate Uptake and Secretion (Metabolomics)”.
Comment 3
One area where the manuscript could be further strengthened is in its consideration of extracellular vesicles (EVs), which are increasingly recognized as important mediators of embryo-maternal communication and carriers of bioactive molecules such as RNAs, proteins, and genomic material. Recent studies on human embryo-derived extracellular vesicles (EVs) from spent culture media have demonstrated their significant potential as non-invasive biomarkers for embryo quality and implantation success. Larger EVs correlated with higher embryo quality, although genomic DNA (gDNA) analysis revealed a high prevalence of chromosomal abnormalities (Veraguas et al., 2020). Comparative analysis indicated that embryos leading to successful pregnancies secreted EVs with higher particle concentration, larger size, and distinct miRNA profiles compared to non-viable embryos, positioning EV content as a promising prognostic tool (Pavani et al., 2022). Furthermore, lower counts of propidium iodide-positive (PI+) EVs in culture media strongly associated with implantation success, reinforcing the diagnostic value of EV nucleic acid content (Pallinger et al., 2017). EV-derived microRNAs (miRNAs) detected in SECM have been associated with pregnancy outcomes (Abu Halima, 2017).
Response
A new section “2.5. Extracellular Vesicles” has been added to characterize this biomarker. All the suggested ideas and references have been included.
Comment 4
It is also worth noting that the segment between lines 47 and 60 presents several generalized claims about non-invasive biomarkers and their clinical application without supporting references. Citing relevant literature here would reinforce the argument and strengthen the manuscript’s academic reliability.
Response
Citations of relevant literature have been added to this manuscript part.
Reviewer 2 Report
Comments and Suggestions for Authors
In the present work, Tesarik try to review noninvasive biomarkers of human embryo developmental potential. In this article, individual markers are reviewed, including marker’s history, current status, as well as available methodologies, strenghts and limitations. However, Figure is limited, and there are some questions that should be explained.
Major concerns
- Some Figures are needed for how to evaluate noninvasive biomarkers of human embryo developmental potential.
- In general, human embryo transfer is performed before blastocyst stage. However, in this review, some biomarkers are detected using a few trophectoderm cells that are from the conceptus in the stage of blastocyst.
- The traditional embryo morphology and kinetics evaluation should be paid more attention.
- Some recent and related papers are not cited. For example,
Porokh V, Kyjovská D, Martonová M, Klenková T, Otevřel P, Kloudová S, Holubcová Z. Zygotic spindle orientation defines cleavage pattern and nuclear status of human embryos. Nat Commun. 2024;15(1):6369.
- Some detail results (including p value) should not be included in this review article. Writing style should be checked and revised throughout the manuscript.
Minor concerns
- Some keywords are not necessary, for example, implantation, pregnancy, and birth.
- Line 32, ‘in vitro’ should be italic.
- Line 35, change ‘(childbirth)’ to ‘pregnancy’.
- Introduction section, the aim of this manuscript should be added.
- Lines 48-61, references are needed.
- Line 64, change ‘in vitro fertilization (IVF)’ to ‘IVF’.
- Lines 219-225, references are needed.
- The Original Images are not necessary.
The English could be improved to more clearly express the research.
Author Response
Dear Reviewer
Thank you very much for your comments which will certainly increase the manuscript quality. I have included the corresponding modifications in the manuscript (highligted). This is a list of the changes made.
Major concerns
- Some Figures are needed for how to evaluate noninvasive biomarkers of human embryo developmental potential.
Response: Two Figures (Figure 1 and Figure 3) have been added. They show, respectively, a schematic representation of sourcing and processing of samples for assessment of non invasive embryo biomarkers in general and that of samples for noninvasive PGT-A in particular.
- In general, human embryo transfer is performed before blastocyst stage. However, in this review, some biomarkers are detected using a few trophectoderm cells that are from the conceptus in the stage of blastocyst.
Response: Embryo transfer is mostly performed on postfertilization day 3 (cleaving embryo) or day 5-6 (blastocyst). Some noninvasive biomarkers can be obtained on day 1, and others on days 3-6. Invasive techniques using trophectoderm biopsy lack precision and can be harmful to further embryo development. This has been made clear in the revised manuscript (highlighted).
- The traditional embryo morphology and kinetics evaluation should be paid more attention.
Response: New data and references concerning the traditional embryo morphology and kinetics evaluation have been added to the Introduction (highlighted).
- Some recent and related papers are not cited. For example,
Porokh V, Kyjovská D, Martonová M, Klenková T, Otevřel P, Kloudová S, Holubcová Z. Zygotic spindle orientation defines cleavage pattern and nuclear status of human embryos. Nat Commun. 2024;15(1):6369.
Response: Several citation of recent papers have been added, including that suggested above.
- Some detail results (including p value) should not be included in this review article. Writing style should be checked and revised throughout the manuscript.
Response: Detailed data including p values have been removed from different parts of the manuscript (struck out and highlighted). Writing style has been checked and revised throughout the manuscript.
Minor concerns
- Some keywords are not necessary, for example, implantation, pregnancy, and birth.
Response: The unnecessary keywords have been removed.
- Line 32, ‘in vitro’ should be italic.
Response: Line 32, the format has been changed to italic.
- Line 35, change ‘(childbirth)’ to ‘pregnancy’.
Response: Line 35, ‘childbirth’ has been changed to ‘pregnancy’.
- Introduction section, the aim of this manuscript should be added.
Response: At the end of the Introduction, the aim of this manuscript has been added.
- Lines 48-61, references are needed.
Response: Lines 48-61, references have been added.
- Line 64, change ‘in vitro fertilization (IVF)’ to ‘IVF’.
Response: Line 64, ‘in vitro fertilization (IVF)’ has been changed to ‘IVF’.
- Lines 219-225, references are needed.
Response: Lines 219-225, references have been added.
- The Original Images are not necessary.
Response: There are no original images in the manuscript. Figure 2 is adapted from Tesarik and Greco, 1999, as acknowledged in the legend.
Round 2
Reviewer 1 Report
Comments and Suggestions for Authors
All reviewer's suggestions and comments have been considered.
Author Response
There are no additional comments by Reviewer 1.
Reviewer 2 Report
Comments and Suggestions for Authors
Thanks for author’s responses. However, there are still some questions that should be explained.
- English grammar and writing style still should be checked and revised throughout the manuscript. For example,
Line 29, change ‘Human Embryo’ to ‘Human’; ‘In Vitro’ should be italic.
- Figure 1. There are some background (triangles), please clean these. In addition, there are some words in capitalization. Day 1, Please move two pronucleuses to the center of embryo.
- Figure 2. There are some background (triangles or others), please clean these. In addition, there are some words in capitalization. Delete ‘(WGA)’. More figure legend is needed.
The English could be improved to more clearly express the research.
Author Response
Dear Reviewer,
Thank you very much for these additional comments which will certainly increase the manuscript quality. I have included the corresponding modifications in the manuscript (highligted). This is a list of the changes made.
English grammar and writing style still should be checked and revised throughout the manuscript. For example,
Line 29, change ‘Human Embryo’ to ‘Human’; ‘In Vitro’ should be italic.
Response: Grammar and style have been rechecked and revised throughout the manuscript and ‘In Vitro’ has been italicized in Line 29.
Figure 1. There are some background (triangles), please clean these. In addition, there are some words in capitalization. Day 1, Please move two pronucleuses to the center of embryo.
Response: The suggested modification has been made.
Figure 2. There are some background (triangles or others), please clean these. In addition, there are some words in capitalization. Delete ‘(WGA)’. More figure legend is needed.
Response: The former Figure 2 is now Figure 3. The suggested modification has been made and the figure legend has been expanded.